# Correlation between Physical Activity and Psychological Distress in Patients Receiving Hemodialysis with Comorbidities: A Cross-Sectional Study

**DOI:** 10.3390/ijerph19073972

**Published:** 2022-03-26

**Authors:** Yu-Hui Wu, Yu-Juei Hsu, Wen-Chii Tzeng

**Affiliations:** 1Graduate Institute of Medical Sciences, National Defense Medical Center, Taipei 11490, Taiwan; nana197926@mail.ndmctsgh.edu.tw; 2Nursing Department, Tri-Service General Hospital, Taipei 11490, Taiwan; 3Nephrology Division, Tri-Service General Hospital, Taipei 11490, Taiwan; yujuei@gmail.com; 4School of Medicine, National Defense Medical Center, Taipei 11490, Taiwan; 5School of Nursing, National Defense Medical Center, Taipei 11490, Taiwan

**Keywords:** hemodialysis, comorbidity, physical activity, depression, anxiety, fatigue

## Abstract

Comorbidities cause psychological distress to patients on hemodialysis and cause their physical function to deteriorate. This study aims to examine whether physical patterns are associated with anxiety, depression and fatigue among patients with and without comorbidities who are on hemodialysis. To this end, a cross-sectional survey was administered to 120 patients on hemodialysis. Data were collected using the International Physical Activity Questionnaire—Short Form, Beck Depression Inventory—Second Edition, Beck Anxiety Inventory, and Brief Fatigue Inventory—Taiwan Version. An independent sample *t* test and generalized linear model analyses were conducted. The results revealed that patients with comorbidities exhibited more severe levels of depression (*p* < 0.001), anxiety (*p* < 0.001), and fatigue (*p* = 0.010) than patients without comorbidities. Additionally, patients on hemodialysis with a high physical activity level (≥600 metabolic equivalent of task per min/week) exhibited less depression (*B* = −4.03; *p* < 0.001; 95% confidence interval [CI] = −6.04, −2.03) and anxiety (*B* = −2.64; *p* = 0.002; 95% CI = −4.27, −1.00) severity than those with a low physical activity level; those who engaged in weekly physical activities exhibited less fatigue severity (*B* = −1.17; *p* = 0.001; 95% CI = −1.84, −0.49) and fatigue interference (*B* = −0.61; *p* = 0.015; 95% CI = −1.10, −0.12). For patients on hemodialysis, having comorbidities was correlated with more severe levels of depression, anxiety, and fatigue. Weekly moderate-intensity physical activities were revealed to be correlated with less severity levels of depression, anxiety, and fatigue. The study findings aid the development of interventions for promoting physical activity among patients on hemodialysis to prevent the exacerbation of complications caused by comorbidities and psychological distress.

## 1. Introduction

Chronic kidney disease (CKD) has an estimated global prevalence of 11.7% to 15.1%; it causes 1.2 million deaths annually and results in 4.902 to 7.083 million patients requiring renal replacement therapy [1]. Comorbidities are major risk factors for morbidity and mortality among patients with CKD [2]. Having a higher number of comorbidities is correlated with an increased risk of death (an increase of 20% to 60% compared with patients with CKD but without comorbidities) [2]. Common comorbidities include hypertension, diabetes, and cardiovascular diseases [3]. When a patient’s comorbidity index score increases during their hemodialysis treatment, they are more likely to experience reduced physical function and activity, increased negative emotions, and increased disease severity [4,5]. Patients on hemodialysis with a greater number of comorbidities have been reported to have more severe depression, higher levels of anxiety [6], and higher fatigue scores [7] than those with fewer comorbidities. Studies have revealed that exercising mitigates anxiety, depression, and fatigue in patients on hemodialysis [8]. However, more evidence is required to verify the role of exercise in relieving the psychological distress of patients on hemodialysis with comorbidities.

Depression, anxiety, and fatigue are common manifestations of psychological distress among patients on hemodialysis [9], and they are all aggravated by the presence of comorbidities, thus increasing the mortality risks of patients [10]. Patients often exhibit symptoms of fatigue after completing a hemodialysis session, [11]. Studies have demonstrated that anemia, dialysis inadequacy, and the presence of comorbidities (e.g., hypertension, diabetes, and cardiovascular disease) may contribute to fatigue [12,13]. Accordingly, patients on hemodialysis may develop daytime sleepiness, depression symptoms, and constant exhaustion [11]. Approximately 42–89% of patients on hemodialysis experience interference in their daily lives because of fatigue symptoms. Therefore, fatigue and psychological distress affect each other in a complex manner [14,15]. The prevalence rate of depression in patients on hemodialysis is 3–4 times higher than that of patients who are not on hemodialysis [16]. This may be because patients on hemodialysis experience CKD-induced physical burdens, a reduced quality of life, and disabilities that affect their physical function [17]. Studies have also reported that anxiety is common among patients on long-term hemodialysis; this population has an anxiety prevalence rate of 43.9%, and their anxiety is often accompanied by depression. Consequently, anxiety may affect patient adherence to hemodialysis treatment and reduce their control over the disease, resulting in adverse outcomes [9].

Insufficient physical activity is a crucial contributing factor to the deterioration of physical function in patients on hemodialysis. Patients who do not engage in physical activity are exposed to increased morbidity risk [18]. Accordingly, the World Health Organization’s 2020 guidelines included updated physical activity–related recommendations for individuals with chronic conditions, anxiety, or depression. The updated recommendations encourage adults to engage in 150–300 min of moderate-intensity physical activities or 75–150 min of high-intensity physical activities weekly to prevent the aggravation of kidney disease or psychological distress [19]; increasing the participation of patients with CKD in daily activities and reducing their CKD-induced burdens are also necessary steps to achieving the objective of “Living Well with Kidney Disease”, which was declared by the World Kidney Day Joint Steering Committee [20]. Studies have also verified the benefits of physical activities for social, psychological, and physiological health [21]; the effectiveness of physical activities in mitigating comorbidities and fatigue and improving physical function [22]; and the effectiveness of physical activities in improving the performance of patients in terms of their dialysis efficiency [23], blood urea nitrogen level (BUN), creatinine level, albumin level, normalized protein catabolic rate (nPCR) [24], and hemoglobin level [25]. Regular physical activity reduces the risk of chronic diseases (e.g., cardiovascular diseases and diabetes), enhances physical function and prevents them from deteriorating with age, and provides overall health benefits [26]. A study that examined patients on hemodialysis after they performed moderate-intensity physical activities reported that those with comorbidities scored less on quality of life than those without comorbidities [27]. Studies have indicated that the mortality rate of patients on hemodialysis with higher levels of physical activity is 52% less than that of patients on hemodialysis with lower levels of physical activity [28]. Fatigue inhibits patients on hemodialysis from actively engaging in physical activities, which leads to these patients developing chronic inflammation that subsequently progresses and causes more fatigue [7]. Accordingly, in the present study, the level of physical activity among patients on hemodialysis with and without comorbidities was compared, and the associations of physical activity with fatigue, anxiety, and depression were explored.

## 2. Materials and Methods

### 2.1. Study Design and Participants

This study utilized a cross-sectional survey design. Structured questionnaires were administered to investigate the physical activity of patients on hemodialysis and to explore the correlation of physical activity with negative emotions. The survey collection period was from January to December 2020, and the research site was the hemodialysis department of a medical center in Taipei. Purposive sampling was conducted to recruit outpatient patients on hemodialysis. Patients were included if they were (1) aged older than 20 years, (2) had regularly received hemodialysis treatment for more than 3 months with three 3-h hemodialysis sessions being held weekly; (3) had sufficient cognitive faculty to complete the questionnaire and express their opinions verbally; (4) could read and communicate in Chinese or Hokkien Taiwanese; and (5) had consented to participate in the research after being informed of its purpose. Patients were excluded if they (1) were diagnosed as having cognitive disabilities or mental illness; (2) could not complete the questionnaire; (3) lacked self-care ability; or (4) were hospitalized and receiving treatment at the time of recruitment. All potential participants were screened by a specialist, and those who met the inclusion criteria were asked to provide their consent to participating in the survey.

G*Power 3.1.9 statistical software was used to determine the sample size required for computing correlation coefficients [29]. A linear multiple regression model F-test was conducted; the effect size (f^2^), significance level, and power were 0.2, 0.05, and 0.80, respectively [30], and the sample size for the present study was 111.

### 2.2. Measurement

#### 2.2.1. Sociodemographic Characteristics

The demographic data of the participants comprised age, gender, education level, marital status, living arrangement, current employment, monthly income, body mass index (BMI, kg/m^2^), existence of chronic diseases (hypertension or diabetes), engagement in regular physical activity (3 times/week), and duration of hemodialysis (years). The biochemical data of the participants included information on dialysis efficiency (Kt/V), nPCR, hemoglobin level (mg/dL), BUN (mg/dL), creatinine level (mg/dL), and albumin level (g/dL) [24].

#### 2.2.2. Charlson Comorbidity Index

The Charlson Comorbidity Index (CCI) assigns weights to 19 diseases on the basis of their association with mortality [31] and it is used to assess the severity of comorbidities [32]. In the CCI, 1 point is assigned if a patient has a history of myocardial infarction, congestive heart failure, peripheral vascular disease, cerebrovascular disease, dementia, chronic pulmonary disease, connective tissue disease, ulcer disease, mild liver disease, or diabetes; 2 points are assigned if a patients has hemiplegia, moderate or severe renal disease, diabetes with end organ damage, any type of tumor, leukemia, or lymphoma; 3 points are assigned if a patient has moderate or severe liver disease; and 6 points are assigned if a patient has a metastatic solid tumor or acquired immunodeficiency syndrome [31]. The CCI had good agreement with a weighted kappa of 0.667 (95% confidence interval [CI] = 0.596–0.714) [33]. The present study used the CCI to measure the comorbidity severity of patients on hemodialysis on the basis of the number of illnesses (more than three) that they had [31]. 

#### 2.2.3. Physical Activity Measure

The present study used the International Physical Activity Questionnaire—Short Form (IPAQ-SF) to measure the physical activity levels of the participants [34]. The questionnaire assessed the participants’ duration of physical activity during the 7 days preceding their filling in of the questionnaire; the intensity and duration of their physical activity (lasting more than 10 min), including low-intensity (≤600 metabolic equivalent of task [MET]-min/week), moderate-intensity (between ≥600 and <3000 MET min/week), and vigorous-intensity (≥3000 MET min/week) physical activity; and their overall physical activity score. Physical activity intensity was measured using MET (kcal/h/kg) units and calculated by multiplying a participant’s weight by the metabolic equivalent of a physical task and the duration of the task [35]. The IPAQ-SF has favorable stability. A Spearman’s rho correlation test revealed that the questionnaire had a test–retest reliability of 0.8 and a criterion validity of 0.30 [36], thus validating its precision and effectiveness.

#### 2.2.4. Depression Measure

The Beck Depression Inventory—Second Edition (BDI-II), which was first proposed by Beck et al. in 1996, was applied to measure the depression severity of the participants. In 2000, The Psychological Corporation published a Chinese translation of the questionnaire. The BDI-II was mainly used to evaluate the depression severity, cognition, and somatic feelings of the participants in the 2 weeks preceding the present study. The questionnaire comprises 21 items, with each item having four options. The participants’ responses for each item are scored on a 4-point scale from 0 to 3 points. The total score ranges from 0–63, with the scores of 0–13, 14–19, 20–28, and 29–63 indicating minimal, mild, moderate, and severe depression, respectively [37]. In the present study, the Cronbach’s alpha for the Chinese version of the BDI-II was 0.92, indicating favorable reliability.

#### 2.2.5. Measurement of Anxiety

To measure anxiety, the present study adopted the Beck Anxiety Inventory (BAI). The scale was first proposed by Beck and Steer in 1987, and a Chinese translation was published by The Psychological Corporation in 2000 [38]. The scale uses 21 self-reporting items, with responses being scored on a 4-point scale (*not at all* = 0; *mildly* = 1; *moderately* = 2; and *severely* = 3). The total scores of 0–9, 10–18, 19–29, and 30–63 indicate minimal, mild, moderate, and severe anxiety, respectively. In the present study, the Cronbach’s alpha for the Chinese version of the BAI was 0.90, indicating acceptable reliability.

#### 2.2.6. Measurement of Fatigue

The Brief Fatigue Inventory—Taiwan version (BFI-T) was employed to assess the fatigue severity of the participants. The scale comprises nine items and is divided into two sections. The first section consists of three items that are used to assess the participant’s current fatigue, usual level of fatigue in the 24 h preceding the filling in of the questionnaire, and worst level of fatigue in the 24 h preceding the filling in of the questionnaire. The mean score of these three items serves as the total score. The second section of the scale has six items that are used to evaluate how fatigue interferes with the lives of respondents in six dimensions, namely general activities, mood, walking ability, normal work (comprising both work outside one’s home and daily chores), relations with other people, and life enjoyment. The mean score of the six items was used as the final score [39]. Each item was scored on a 11-point Likert scale from 0–10 points. For the first section, scores of 0, 1–4, 5–6, and 7–10 indicate no, mild, moderate, and severe fatigue, respectively. For the second section, higher scores indicate greater interference; specifically, scores of 0, 1–4, 5–6, and 7–10 indicate no, mild, moderate, and severe fatigue, respectively, and a corresponding severity of interference [40]. In the present study, the Cronbach’s alpha values for the Chinese version of the BFI-T were 0.90, and 0.92 for fatigue severity and fatigue inference, respectively.

### 2.3. Ethical Considerations

In the present study, participants were enrolled following the approval of the study by the relevant institutional review board (IRB number: 1-108-05-195). The participants participated voluntarily in the present study, and they were not required to bear any additional cost. Furthermore, they could withdraw from the study at any time without affecting their right to receive medical care in the future. A researcher provided a thorough explanation of the study to the participants and obtained their consent to participate prior to their enrolment in the study.

### 2.4. Data Analysis

Statistical analysis was performed using SPSS version 22.0 (SPSS, Chicago, IL, USA), and the level of significance was set to 0.05. The descriptive statistical data that were obtained comprised the participants’ sociodemographic data and physical activity, anxiety, depression, and fatigue scores, all of which were expressed as numbers (%), means, standard deviations (SDs), or ranges. Subsequently, inferential statistics were calculated using an independent sample *t* test and generalized linear modeling. Univariate linear regression was performed to verify the correlations among the examined comorbidities; sociodemographic characteristics; and physical activity, depression, anxiety, and fatigue scores. Multivariate analyses that incorporated linear regression models were conducted, and adjustments were made for sociodemographic characteristics (age, gender, education level, marital status, employment, monthly income, Kt/V, nPCR, hemoglobin level, BUN, creatinine level, albumin level, regular physical activity, duration of hemodialysis, and comorbidities) and physical activity. A generalized linear model was used to identify key explanatory variables associated with the physical activity, anxiety, depression, and fatigue domains. This method provides flexible statistical modeling and allows for the nonnormal distributions of dependent variables [41].

## 3. Results

### 3.1. Patient Demographics

During the survey collection period, 136 patients on hemodialysis consented to participating in the survey. However, five participants subsequently refused to complete the questionnaire, and 11 lacked the illiteracy level that was required to read and understand the questionnaire. After the responses of the aforementioned participants were excluded, 120 patients on hemodialysis completed the questionnaire.

The demographic characteristics of the 120 patients on hemodialysis are presented in Table 1. The mean age (SD) of the participants was 61.46 years (10.32). Most participants were men (93; 77.5%), had at least a senior high school education (75; 62.5%), were married (79; 65.8%), and lived with their family (107; 89.2%). Most of them were unemployed (98; 81.7%) and had a monthly income of less than NT$50,000 (108; 90%). They had a mean BMI (SD) score of 23.74 kg/m^2^ (4.10), mean Kt/V score (SD) of 1.49 (0.31), mean nPCR score of 1.16 g/kg/d (0.27), mean BUN score (SD) of 68.08 mg/dL (18.97), mean creatinine level score (SD) of 9.73 mg/dL (2.23), and mean albumin score (SD) of 3.92 g/dL (0.33). The mean duration of their CKD (SD) following diagnosis was 12.37 years (4.52). The mean duration of their hemodialysis (SD) was 5.68 years (4.37). Among the participants, a CCI-based examination revealed that 59 patients (49.2%) had comorbidities, with the most common comorbidity being hypertension (93; 77.5%); the other comorbidities (presented in descending order of prevalence) were diabetes (57; 47.5%), peripheral vascular disease (51; 42.5%), congestive heart failure (41; 34.2%), and others (24; 19.9%). Sixty-one participants (50.8%) engaged in regular physical activity, and only 54 (45%) reported sufficient energy expenditure (≥600 MET/week). The participants’ mean BDI-II (SD) and mean BAI (SD) scores were 5.59 (5.96) and 3.10 (4.74), respectively. The participants’ mean fatigue severity and mean fatigue interference scores on the BFI-T were 3.58 (1.98) and 2.27 (1.41), respectively.

### 3.2. Depression, Anxiety, and Fatigue

The participants had a mean BDI-II score of 5.59 (SD = 5.96, range = 0.0–38.0); eight participants (6.7%) had a score of ≥14. Their mean BAI score was 3.10 (SD = 4.74, range = 0.0–21.0); 13 participants (10.9%) had a score of ≥10. Their mean BFI-T scores for fatigue severity and fatigue inference were 3.58 (SD = 1.98) and 2.27 (SD = 1.41), respectively. Their BDI-II scores were positively correlated with anxiety (*r* = 0.69, *p* < 0.001), fatigue severity (*r* = 0.02, *p* = 0.777), and fatigue inference (*r* = 0.01, *p* = 0.906). No multicollinearity was detected.

### 3.3. Regression Analysis

Table 2 summarizes the results of a univariate linear regression analysis. For BDI-II scores, age (*B* = 0.14, *p* = 0.006, 95% CI [0.04, 7.69]), gender (*B* = 3.29; *p* = 0.009; 95% CI = 0.82, 5.77), having more than 12 years of education (*B* = −3.14; *p* = 0.004; 95% CI = −5.26, −1.02), being currently employed (*B* = −3.84; *p* = 0.005; 95% CI = −6.40, −1.18), having a monthly income of more than NT$50,000 (*B* = −3.62; *p* = 0.041; 95% CI = −7.10, −0.14), increased dialysis efficiency (*B* = −4.28; *p* = 0.011; 95% CI = −7.58, −0.98), hemodialysis duration (*B* = −0.28; *p* = 0.021; 95% CI = −0.52, −0.04), and a weekly energy expenditure of ≥600 MET (*B* = −4.03; *p* < 0.001; 95% CI = −6.04, −2.03) were negatively correlated.

The scores were subsequently analyzed in a multivariate analysis to adjust for factors pertaining to sociodemographic characteristics and physical activity. Significant differences were observed between patients on hemodialysis with and without comorbidities in terms of their depression (*p* < 0.001), anxiety (*p* < 0.001), and fatigue severity (*p* = 0.010) scores. No significant difference was detected between the two groups for fatigue interference. Furthermore, significant differences between patients on hemodialysis with and without an exercise capacity of ≥600 MET were observed in terms of their depression (*p* < 0.001), anxiety (*p* < 0.001), and fatigue severity (*p* = 0.010) scores. Between the patients with and without an exercise capacity of ≥600 MET, no significant difference in fatigue interference was detected (Table 1, Table 2 and Table 3).

Table 3 lists the results of a univariate linear regression analysis. For BAI scores (3.10, SD *=* 4.74), female gender (*B* = 2.93; *p* = 0.003; 95% CI = 0.97, 4.88), having more than 12 years of education (*B* = −3.14; *p* < 0.001; 95% CI = −4.83, −1.53), being currently employed (*B* = −3.18; *p* = 0.003; 95% CI = −5.29, −1.07), having a monthly income of more than NT$50,000 (*B* = −2.88; *p* = 0.041; 95% CI = −5.65, −0.11), increased dialysis efficiency (*B* = −5.73; *p* < 0.001; 95% CI = −8.23, −3.24), hemodialysis duration (*B* = −0.26; *p* = 0.006; 95% CI = −0.45, −0.07), and a weekly energy expenditure of ≥600 MET (*B* = −2.64; *p* = 0.002; 95% CI = −4.27, −1.00) were negatively correlated.

Table 4 presents the results of a univariate linear regression analysis. For BFI-T scores, fatigue severity (3.58, SD = 1.98), increased dialysis efficiency (*B* = 1.89; *p* = 0.001; 95% CI = 0.82, 2.97), and regular weekly physical activity (*B* = −1.17; *p* = 0.001; 95% CI = −1.84, −0.49) were negatively correlated, whereas Kt/V (*B* = 1.89; *p* = 0.001; 95% CI = 0.82, 2.97) was positively correlated. For the fatigue interference scores (2.27, SD = 1.41) of the BFI-T, increased dialysis efficiency (*B* = 1.13; *p* = 0.004; 95% CI = 0.35, 1.90) and regular weekly physical activity (*B* = −0.61; *p* = 0.015; 95% CI = −1.10, −0.12) were negatively correlated, whereas Kt/V (*B* = 1.13; *p* = 0.004; 95% CI = 0.35, 1.90) was positively correlated.

Among the patients with comorbidities, those who engaged in moder-ate-intensity physical activity had significantly lower overall BDI-II (*p* = 0.032), and BAI (*p* = 0.029). However, both fatigue severity and fatigue interference scores did not differ significantly by comorbidities (Figure 1).

## 4. Discussion

For the patients on hemodialysis with comorbidities, significant differences in depression, anxiety, and fatigue severity scores were detected between those who engaged in moderate-intensity weekly physical activities and those who engaged in low-intensity weekly physical activities. Furthermore, regular physical activity was revealed to be negatively correlated with fatigue severity and fatigue interference. The depression, anxiety, and fatigue severity levels of the patients on hemodialysis who engaged in moderate-intensity weekly physical activities were significantly less than those of the patients who engaged in low-intensity weekly physical activities. However, no significant differences between the two groups were observed for fatigue interference.

Our results revealed that comorbidities were correlated with physical activity capacity, depression severity, anxiety severity, and fatigue severity and that regular physical activity was correlated with fatigue severity. A lack of physical activity was identified as the main factor influencing the deterioration of physical function in patients on hemodialysis, which in turn increases their morbidity risk [18]. Physical activity has positive health effects in patients with CKD, and it is a feasible adjuvant treatment for mitigating morbidity risk and disease-induced comorbidities. Accordingly, exercise and physical activity are crucial treatments and preventative measures for diseases. The concept of “exercise as medicine” emphasizes that exercise not only helps to enhance physical function and muscle strength but also provides the positive benefit of slowing chronic disease progression [42]. The literature has reported that insufficient physical activity is the main factor that causes the loss of physical function, reduction of exercise capacity, and development of muscular dystrophy [18]. Comorbidities and diseases share a complex association and mutually influence each other; both affect the physical function and survival of a patient, with comorbidities increasing the severity of a disease [43]. Implementing physical activity programs for patients significantly improves their physical function and mental health. Studies have revealed that engagement in regular moderate-intensity physical activities prevents coronary heart disease and type 2 diabetes and reduces mortality, anxiety, and depression symptoms in patients on hemodialysis [19]. Engaging in regular physical activity provides various health benefits, including a reduced risk of chronic diseases, enhanced physical function, and a slower decline of physical function with age [26].

The present study further demonstrated that the physical activity of patients on hemodialysis with comorbidities was correlated with their depression and anxiety severity; in particular, moderate-intensity physical activities had a significant effect on their comorbidities. These results may be attributable to anxiety and depression negatively influencing intention to engage in physical activity in comorbidity patients, resulting in a cycle in which physical inactivity contributed to the patients’ anxiety and depression. This finding should be further explored to improve psychological conditions in patients on hemodialysis with comorbidities.

Patients on hemodialysis have a higher risk of depression when their capacity to engage in physical activity is reduced [5]. This may be related to physical burdens, quality of life, and role loss of such patients because of their chronic disease [18]. In addition to providing pharmaceutical therapy, clinical care personnel must consider the effect of physical activity on the depression status of patients. Exercise is beneficial for treating depression symptoms. The incorporation of physical activity interventions in hemodialysis treatment can effectively reduce the depression of patients on hemodialysis, which has a positive effect of their disease treatment [44]. Exercise also stimulates several neuroplastic processes that are implicated in depression, mitigates chronic inflammation, induces the release of endorphins to increase stress resilience, and increases the self-esteem and self-efficacy of individuals [45]. Exercise interventions, including aerobic, resilience, and muscle training, can ameliorate depression symptoms [46]. Therefore, physical activity can serve as a nonpharmaceutical intervention for patients on dialysis, prevent severe comorbidities from developing during hemodialysis, and prevent the development of depression-related comorbidities [47]. A Taiwan-based study reported that after adjustments were made for sociodemographic factors and medical comorbidities, a correlation was detected between comorbidities and depression symptoms; this finding is consistent with that of the present study [48].

The present study revealed that engagement in moderate-intensity physical activity was significantly correlated with a reduction in anxiety among patients on hemodialysis; this finding is similar to that reported by Dziubek et al. (2016) [31]. This phenomenon occurs because physical activity induces the structural reconstruction of the hippocampus, amygdala, and prefrontal cortex [49]. Physical activity training programs can serve as a nonpharmaceutical treatment that can increase the positive emotions of patients on hemodialysis, improve their health and motivation, and alter their behavior. Accordingly, weekly physical activity programs can reduce the anxiety of patients on hemodialysis and improve their physical function [47]. Health care personnel should guide patients on hemodialysis to engage in regular physical activity and implement training programs that help to mitigate their anxiety.

In our study, patients engaged in three 30-min physical activity sessions per week. A significant correlation was observed between the fatigue severity and fatigue interference scores of all patients on hemodialysis, but no significant correlation between fatigue severity and fatigue interference scores was observed among the patients on hemodialysis who engaged in regular moderate-intensity physical activities. This could be because the patients on hemolysis all experienced general fatigue. Accordingly, future studies that focus on physical activity interventions should first direct patients on hemodialysis to engage in low-intensity physical activities and then increase the intensity of such activities in accordance with physical conditions of the patients. A study that comprised a systematic review and meta-analysis reported that fatigue is more common in patients on hemodialysis than in those who are not on hemodialysis. At the time of writing, fatigue cannot be prevented or controlled using medicine. However, studies have revealed that regular physical activity improves the functions of the neuroendocrine system (e.g., the hypothalamic–pituitary–adrenal axis) and increases muscle catabolism [50,51], thereby significantly reducing fatigue in patients on hemodialysis. Additionally, patients on hemodialysis who engage in various types of moderate-intensity training programs (e.g., aerobic and resilience training) regularly experienced reduced side effects of medication [52,53].

Dialysis efficiency, as measured using the Kt/V score, is a suitable indicator for measuring waste removal efficiency during hemodialysis, and it can be used to minimize hemodialysis-related disease and morbidity risks. A higher Kt/V score indicates that patients can maintain a more favorable clinical condition after hemodialysis, be more energetic, maintain a regular blood volume and blood pressure, and experience a higher quality of life [54]. The study reported that a higher mean Kt/V score is correlated with a lower depression score [55], and that Kt/V is positively correlated with fatigue [11]. Patients on hemodialysis with comorbidities often experience fatigue, and a positive correlation between exercising for less than 1 h daily and fatigue was identified [56]. However, few studies have explored whether Kt/V scores and anxiety are correlated [57]. Hornik et al. [58] reported that relative to patients on hemodialysis who engaged in a low level of physical activity on an irregular basis, those who engaged in a high level of physical activity on a regular basis have a lower comorbidity rate; their study also posited that regular physical activity improves the effect of hemodialysis on patients. Patients on hemodialysis require an effective method for preventing disease exacerbation and reducing morbidity. Studies have verified that 30 min of moderate-to-high-intensity physical activity daily or 150 min of moderate-to-high-intensity physical activity weekly significantly improves the physical strength, energy level, self-care ability, and Kt/V scores of patients on hemodialysis; these findings suggest that physical activity enhances the effect of hemodialysis [23,59], and they are consistent with those of the present study. 

The literature has indicated that fatigue is correlated with age, gender, disease duration, cognitive disabilities, and anxiety and depression symptoms. Patients who are older, women, have a longer disease duration, have cognitive disabilities, and were apathetic, anxious, and depressed experience higher levels of fatigue [60]. Moreover, hemodialysis duration is significantly correlated with anxiety, depression, and fatigue [31,61]. Additionally, studies have revealed that marital status, having a fixed income, having chronic diseases, and lack of physical activity are correlated with depression in patients on hemodialysis [47]. BMI and physical activity status of patients are correlated with receiving more than 12 years of education, having a fixed income, being female, and being employed [62]. Furthermore, the physical activity status of patients on hemodialysis can be determined by assessing whether they have a high educational level, hypertension, or cardiovascular disease; their reason for receiving hemodialysis; and the condition of their musculoskeletal system [63]. Research has yet to reveal any significant differences in physical activity among participants with different sociodemographic variables, including age, gender, educational level, weight, and BMI. Patients on hemodialysis who participated in physical activity interventions exhibited significantly less anxiety and fewer depression symptoms than those who did not. Relatedly, a study indicated that patients on hemodialysis who ceased to engage in physical activities had significantly higher anxiety and depression scores [64], and it argued that patients on hemodialysis engage in regular physical activity to maintain their mental health; the finding of that study is consistent with those of the present study.

The limitations of the present study, specifically its small sample size, limited the generalizability of its research results. Therefore, future studies should recruit participants from large sample pools or collaborate with multiple medical centers to increase the generalizability of their results. Furthermore, the present study’s cross-sectional research design precluded causal inference. Nevertheless, the present study provides preliminary evidence for various relationships pertaining to patients on hemodialysis, comorbidities, physical activity, depression, anxiety, and fatigue. However, the present study did not include psychological distress–related variables (e.g., sleep disorders) in its analysis. Future studies should expand on the questionnaire content and measure physical activity to further explore the correlation between physical activity and negative emotions in patients on hemodialysis with comorbidities.

Future studies should explore early interventions for patients on hemodialysis, specifically those involving weekly training programs with moderate-intensity physical activities that effectively prevent disease exacerbation and negative emotions resulting from comorbidities. The present study provides preliminary evidence supporting the implementation of weekly physical activity training sessions of moderate intensity to improve the physical health of patients on hemodialysis with comorbidities. Fatigue can prevent patients from engaging in physical activity, leading to increased depression and anxiety. In this regard, regular physical activity can mitigate their fatigue, depression, and anxiety, thereby improving their physical and mental health.

## 5. Conclusions

Although the participants examined in the present study reported that they exercised regularly, they could have been unable to achieve the intended effects because the intensity level of their physical activity was insufficient. Our results revealed that patients must perform moderately intense physical activities to reduce their depression, anxiety, and fatigue. The correlation of moderate-intensity physical activity with depression, anxiety, and fatigue can serve as a reference for medical professionals who are providing care to patients on hemodialysis with comorbidities. Finally, the present study provides a foundation for developing appropriate interventions that reduce fatigue, anxiety, and depression in patients on hemodialysis.

## Figures and Tables

**Figure 1 ijerph-19-03972-f001:**
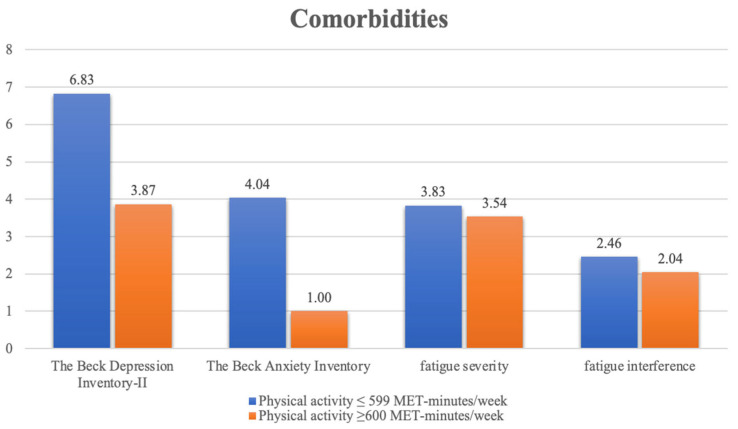
Comparison of physical activity and psychological distress of patients on hemodialysis.

**Table 1 ijerph-19-03972-t001:** Sociodemographic characteristics of patients on hemodialysis (*n* = 120).

Variables	*n*	%	Mean (SD)
Age (years)			
≤59	45	37.5	3.75 (3.11)
≥60	75	62.5	6.69 (6.93)
Gender			
Male	93	77.5	4.84 (4.45)
Female	27	22.5	8.14 (9.14)
Education			
elementary/Junior high School	45	37.5	7.55 (7.66)
Senior high School or above	75	62.5	4.41 (4.29)
Marital status			
Unmarried	41	34.2	5.82 (6.94)
Married	79	65.8	5.46 (5.42)
Living arrangement			
Alone	13	10.8	4.15 (3.41)
with family	107	89.2	5.76 (6.18)
Current employment			
Yes	22	18.3	2.45 (2.42)
No	98	81.7	6.29 (6.29)
Monthly income/(NTD)			
<50,000	108	90.0	5.95 (6.14)
≥50,000	12	10.0	2.33 (2.10)
BMI, kg/m^2^			
<25	79	65.8	5.93 (6.80)
25–29.9	31	25.8	4.83 (3.09)
≥30	10	8.4	5.20 (5.78)
Biochemical data			
Kt/V			
nPCR (g/kg/d)			
Hemoglobin (mg/dL)			
Blood urea nitrogen (BUN; mg/dL)			
Creatinine (mg/dL)			
Albumin (g/dL)			
Comorbidities			
Yes	59	49.2	6.03 (5.20)
No	61	50.8	5.16 (6.62)
Comorbid conditions			
Hypertension	93	77.5	
Diabetes	57	47.5	
Peripheral vascular disease	51	42.5	
Congestive heart failure	41	34.2	
others	24	19.9	
Duration of CKD (years)			12.37 (4.52)
Regular Physical activity (3 times/week)			
Yes	61	50.8	5.08 (4.72)
No	59	49.2	6.11 (7.01)
Duration of hemodialysis (years)			5.68 (4.37)
Physical activity (MET-min/week)			
≥600 MET-min/week	54	45	3.37 (3.31)
≤599 MET-min/week	66	55	7.40 (6.97)
BDI-II			5.59 (5.96)
BAI			3.10 (4.74)
BFI-Taiwan version			
fatigue severity			3.58 (1.98)
fatigue interference			2.27 (1.41)

Abbreviations: NTD, New Taiwan Dollar; BMI, body mass index; nPCR, normalized protein catabolic rate; BDI-II, Beck Depression Inventory-Second Edition; BAI, Beck Anxiety Inventory; BFI-T, Brief Fatigue Inventory-Taiwan version. Presence of comorbidities was defined as having a Charlson Comorbidity Index score of ≥3. Physical activity was measured using the International Physical Activity Questionnaire-Short Form.

**Table 2 ijerph-19-03972-t002:** Analysis of factors associated with Beck Depression Inventory-Second Edition score and sociodemographic characteristics (*n* = 120).

The Beck Depression Inventory-Second Edition					
		95% CI			
Variables	*B*	Lower	Upper	*p*	*p^a^*
Age (years)	0.14	0.04	7.69	0.006	
≤59					
≥60					
Gender	3.29	0.82	5.77	0.009	
Male					
Female					
Education	−3.14	−5.26	−1.02	0.004	
elementary/Junior high School					
Senior high School or above					
Marital status	−0.36	−2.60	1.87	0.752	
Unmarried					
Married					
Living arrangement	−0.77	−2.17	5.16	0.641	
Alone					
with family					
Current employment	−3.84	−6.40	−1.18	0.005	
Yes					
No					
Monthly income/(NTD)	−3.62	−7.10	−0.14	0.041	
<50,000					
≥50,000					
BMI, kg/m^2^	0.01	−0.24	0.27	0.907	
<25					
25–29.9					
≥30					
Biochemical data					
Kt/V	−4.28	−7.58	−0.98	0.011	
nPCR (g/kg/d)	−0.74	−4.26	2.77	0.679	
Hemoglobin (mg/dL)	−0.29	−1.03	0.44	0.437	
Blood urea nitrogen (BUN; mg/dL)	0.00	−0.05	0.06	0.819	
Creatinine (mg/dL)	0.20	−0.26	0.68	0.392	
Albumin (g/dL)	−1.57	−4.76	1.60	0.331	
Comorbidities	0.87	−1.24	2.98	0.421	<0.001
Yes					
No					
Regular Physical activity (3 times/week)	−1.03	−3.15	1.08	0.337	
Yes					
No					
Duration of hemodialysis (years)	−0.28	−0.52	−0.04	0.021	
Physical activity (MET-min/week)	−4.03	−6.04	−2.03	<0.001	<0.001
≥600 MET-min/week					
≤599 MET-min/week					

Abbreviations: NTD, New Taiwan Dollar; BMI, body mass index; nPCR, normalized protein catabolic rate; Note: Physical activity was measured using the International Physical Activity Questionnaire-Short Form. Presence of comorbidities was defined as having a Charlson Comorbidity Index score of ≥3. *p* values were obtained using univariate analysis that was based on linear regression models. *p^a^* values were obtained from a multivariate linear regression model, with adjustments being made for sociodemographic characteristics (i.e., age, gender, education, marital status, employment, monthly income, Kt/V, nPCR, hemoglobin level, blood urea nitrogen, creatinine level, albumin level, regular physical activity, duration of hemodialysis, and comorbidities) and physical activity.

**Table 3 ijerph-19-03972-t003:** Analysis of factors associated with Beck Anxiety Inventory score and sociodemographic characteristics (*n* = 120).

The Beck Anxiety Inventory					
		95% CI			
Variables	*B*	Lower	Upper	*p*	*p^a^*
Age (years)	0.07	−0.00	0.16	0.055	
≤59					
≥60					
Gender	2.93	0.97	4.88	0.003	
Male					
Female					
Education	−3.18	−4.83	−1.53	<0.001	
elementary/Junior high School					
Senior high School or above					
Marital status	−0.51	−2.29	1.26	0.571	
Unmarried					
Married					
Living arrangement	−2.18	−0.59	4.95	0.496	
Alone					
with family					
Current employment	−3.18	−5.29	−1.07	0.003	
Yes					
No					
Monthly income/(NTD)	−2.88	−5.65	−0.11	0.041	
<50,000					
≥50,000					
BMI, kg/m^2^	0.09	−0.10	0.30	0.348	
<25					
25–29.9					
≥30					
Biochemical data					
Kt/V	−5.73	−8.23	−3.24	<0.001	
nPCR (g/kg/d)	−1.15	−3.94	1.64	0.420	
Hemoglobin (mg/dL)	−0.06	−0.66	0.52	0.819	
Blood urea nitrogen (BUN; mg/dL)	0.03	−0.01	0.07	0.166	
Creatinine (mg/dL)	0.16	−0.21	0.54	0.385	
Albumin (g/dL)	−0.48	−3.03	2.05	0.707	
Comorbidities	0.23	−1.45	1.92	0.789	<0.001
Yes					
No					
Regular Physical activity (3 times/week)	0.70	−0.88	2.48	0.354	
Yes					
No					
Duration of hemodialysis (years)	−0.26	−0.45	−0.07	0.006	
Physical activity (MET-min/week)	−2.64	−4.27	−1.00	0.002	<0.001
≥600 MET-min/week					
≤599 MET-min/week					

Abbreviations: NTD, New Taiwan Dollar; BMI, body mass index; nPCR, normalized protein catabolic rate. Note: Physical activity was measured using the International Physical Activity Questionnaire-Short Form. Presence of comorbidities was defined as having a Charlson Comorbidity Index score of ≥3. *p* values were obtained using univariate analysis that was based on linear regression models. *P^a^* values were obtained from a multivariate linear model, with adjustment being made for sociodemographic characteristics (age, gender, education, marital status, employment, monthly income, Kt/V, nPCR, hemoglobin level, blood urea nitrogen, creatinine level, albumin level, regular physical activity, duration of hemodialysis, and comorbidities) and physical activity.

**Table 4 ijerph-19-03972-t004:** Analysis of factors associated with Brief Fatigue Inventory-Taiwan version scores and sociodemographic characteristics (*n* = 120).

The Brief Fatigue Inventory-Taiwan Version										
	Fatigue Severity					Fatigue Interference				
		95% CI					95% CI			
Variables	*B*	Lower	Upper	*p*	*p^a^*	*B*	Lower	Upper	*p*	*p^a^*
Age (years)	0.01	−0.01	0.05	0.387		−0.00	−0.02	0.02	0.960	
≤59										
≥60										
Gender	0.62	−0.22	1.46	0.148		0.52	−0.07	1.11	0.087	
Male										
Female										
Education	−0.26	−0.99	0.46	0.472		−0.15	−0.67	0.36	0.555	
elementary/Junior high School										
Senior high School or above										
Marital status	0.12	−0.61	0.87	0.736		−0.08	−0.61	0.45	0.767	
Unmarried										
Married										
Living arrangement	0.81	−0.40	2.03	0.385		0.10	−2.00	2.21	0.605	
Alone										
with family										
Current employment	0.12	−0.79	1.03	0.791		0.25	−0.39	0.90	0.441	
Yes										
No										
Monthly income/(NTD)	0.37	−0.80	1.55	0.535		0.24	−0.58	1.08	0.561	
<50,000										
≥50,000										
BMI, kg/m^2^	−0.02	−0.11	0.06	0.571		−0.03	−0.10	0.02	0.215	
<25										
25–29.9										
≥30										
Biochemical data										
Kt/V	1.89	0.82	2.97	0.001		1.13	0.35	1.90	0.004	
nPCR (g/kg/d)	0.56	−0.60	1.73	0.340		0.16	−0.67	0.99	0.703	
Hemoglobin (mg/dL)	−0.09	−0.34	0.14	0.433		0.01	−0.16	0.19	0.870	
Blood urea nitrogen (BUN; mg/dL)	−0.01	−0.02	0.00	0.293		0.00	−0.01	0.01	0.682	
Creatinine (mg/dL)	0.05	−0.10	0.21	0.471		0.10	−0.00	0.21	0.069	
Albumin (g/dL)	−0.50	−1.56	0.56	0.355		−0.14	−0.90	0.61	0.705	
Comorbidities	0.34	−0.35	1.05	0.335	0.010	0.14	−0.36	0.64	0.586	0.211
Yes										
No										
Regular Physical activity (3 times/week)	−1.17	−1.84	−0.49	0.001		−0.61	−1.10	−0.12	0.015	
Yes										
No										
Duration of hemodialysis (years)	0.07	−0.11	0.15	0.089		0.04	−0.00	0.10	0.100	
Physical activity (MET-min/week)	−0.04	−0.75	0.66	0.900	0.010	−0.14	−0.64	0.36	0.578	0.211
≥600 MET-min/week										
≤599 MET-min/week										

Abbreviations: NTD, New Taiwan Dollar; BMI, body mass index; nPCR, normalized protein catabolic rate. Note: Physical activity was measured using the International Physical Activity Questionnaire-Short Form. Presence of comorbidities was defined as having a Charlson Comorbidity Index score of ≥3. The *p* values were obtained using univariate analysis that was based on linear regression models. *p^a^* values were obtained using multivariate analysis that was based on linear regression models, with adjustment being made for sociodemographic characteristics (age, gender, education, marital status, employment, monthly income, Kt/V, nPCR, hemoglobin level, blood urea nitrogen, creatinine level, albumin level, regular physical activity, duration of hemodialysis, and comorbidities) and physical activity.

## Data Availability

The datasets used and/or analyzed during the current study are available from the corresponding author on reasonable request.

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
