# Peer review of "Correlation between Physical Activity and Psychological Distress in Patients Receiving Hemodialysis with Comorbidities: A Cross-Sectional Study"

_ijerph, 2022, doi:10.3390/ijerph19073972_

Round 1

Reviewer 1 Report

This study explored the effect of physical activity on depression, anxiety, and fatigue in patients receiving hemodialysis with comorbidities. The title and abstract cover the main aspect of the work. The introduction provides background and information relevant to the study. The abstract is somehow confusing at the beginning as it is stated that physical activity is correlated with different factors....then, the paper aims to explore the correlation between the same elements. "Physical activity is correlated with psychological distress, including depression, fatigue, and anxiety, in patients receiving hemodialysis, particularly those with comorbidities. To explore the correlation of physical activity level with fatigue, anxiety, and depression in patients receiving hemodialysis with and without comorbidities,". I suggest rephrasing the beginning of the abstract.

The methods are clear and replicable. Moreover, the results match the methods described. The findings related by the author are novel and bring an advance in the field.

Reviewer 2 Report

Introduction: please provide gap of knowledge why the authors focused on the comorbidities. 

Method: Please provide the information of the Charlson Comorbidity Index; reliability or validity and the disease of those. 

Results: please provide the disease of comorbidities and duration of disease and analyses. 

Conclusion: Please provide the reasons why regular physical activity was not associated with anxiety or depression but physical activity was related. 

Please also provide the discussion part in term of specific disease e.g, DM, HT that might be related to PA or psychological distress.

Reviewer 3 Report

Review for the manuscript entitled “Correlation Between Physical Activity and Psychological Distress in Patients Receiving Hemodialysis with Comorbidities: A Cross-Sectional Study”.

This manuscript presents the results of a cross-sectional study investigating the correlation between physical activity and psychological distress in hemodialysis patients. The results are reasonable: high physical activity is generally associated with decreased psychological distress. This result is consistent with the study published recently by the authors (Int J Environ Res Public Health. 2022 Jan 12;19(2):811).

However, it is unclear what the new findings are in this study. And the presentation and interpretation of the results is often unclear.

The concerns are as follows:

  1. The relationship between physical inactivity and psychological distress in hemodialysis patients seems to be well documented. What's new in this manuscript?
  2. Since this is a cross-sectional study, the relationship between physical activity and psychological distress is not necessarily causal. For example, if psychological distress is upstream of causation, then inappropriate promotion of physical activity may have the opposite effect.
  3. The bulk of the Discussion is a review of previous reports on physical activity and psychological distress, rather than a discussion of the results of the current study.
  4. Non-parametric factors should not be specified as Mean±SD.
  5. The methods of statistical analysis are not sufficiently described. For example, what type of univariate linear regression was used?
  6. In all tables, why is linear regression used instead of between-groups significance testing, despite the fact that the subjects are grouped?
  7. What is the meaning of "comorbidity"? For example, what does it mean that the Mean (SD) of subjects with ≥600 METs/week in Table 4 is 1929.87 (1790.96)?

Reviewer 4 Report

This study analyzed the effect of physical activity and comorbidities on depression, anxiety, and fatigue in hemodialysis patients. Previously the authors reported on physical functioning but the article was not cited. I assume it is not separate work so it should be mentioned.

The problem of disease burden and low quality of life is not new and was extensively studied in patients with kidney disease. The year 2021 was declared the year of “Living Well with Kidney Disease” with the aim of reduction of the burden and consequences of CKD-related symptoms to enable life participation. 

Physical activity was used previously in various clinical trials as a intervention to improve patient outcome. The review done by Hargrove suggests that in adults on maintenance hemodialysis, aerobic exercise improves several hemodialysis-related symptoms, including restless legs syndrome, symptoms of depression, muscle cramping, and fatigue. Recently,  metanalysis to assess the benefits and safety of regular structured exercise training in adults undergoing dialysis on patient-important outcomes including death, cardiovascular events, fatigue, functional capacity, pain, and depression was published. 

It raises the question whether another cross-sectional study is necessary? It is my major concern. According to the manuscript, the introduction is too long. The study group was not presented properly. It is mentioned that it was shown in Table 1, but it contains Beck Depression Inventory. Multivariate analyses are shown in unclear way, only p value is shown. Direction of correlation should verified in the text (positive and negative B, but information about negative correlation). Why linear regression was used to assess physical activity effect on comorbidities? Comorbidity is presented as dichotomous value.

Round 2

Reviewer 2 Report

The manuscript has been accepted for publication. 

Author Response

Thank you for the positive feedback.

Reviewer 3 Report

Review for the revised manuscript.

The quantification of comorbidity by the addition of The Charlson Comorbidity Index is a favorable modification.

On the other hand, further revises are needed in the following points.

>(Results) “In addition, among the patients with comorbidities, those who engaged in moderate-intensity physical activity had significantly higher overall BDI-II (p = 0.032), BAI (p = 0.029), fatigue severity (p = 0.611), and fatigue interference (p = 0.326) scores than those who engaged in low-intensity physical activity (Figure 1).”

For fatigue severity and interference score, it is not significantly higher.

> (Discussion) “Therefore, patients on hemodialysis with comorbidities should be actively encouraged to participate in moderate-intensity physical activity training, which can improve their mental and physical health and mitigate comorbidity-related negative effects.

The authors repeatedly emphasized that physical activity training improves mental health. However, this study is a cross-sectional study, but not an intervention study. Therefore, it is not possible to conclude this from the present results.

Correlation and causation are different. It is both possible that low activity promotes depression and, conversely, that depression suppresses activity.

Author Response

Point 1: The quantification of comorbidity by the addition of The Charlson Comorbidity Index is a favorable modification.

Response 1: Thank you again for you time and positive feedback.

Point 2: (Results) “In addition, among the patients with comorbidities, those who engaged in moderate-intensity physical activity had significantly higher overall BDI-II (p = 0.032), BAI (p = 0.029), fatigue severity (p = 0.611), and fatigue interference (p = 0.326) scores than those who engaged in low-intensity physical activity (Figure 1).” For fatigue severity and interference score, it is not significantly higher.

Response 2: As suggested, we have rephrased the results on page 12 as follows:

“Among the patients with comorbidities, those who engaged in moderate-intensity physical activity had significantly lower overall BDI-II (p = 0.032), and BAI (p = 0.029). However, both fatigue severity and fatigue interference scores did not differ significantly by comorbidities (Figure 1).”

Point 3: (Discussion) “Therefore, patients on hemodialysis with comorbidities should be actively encouraged to participate in moderate-intensity physical activity training, which can improve their mental and physical health and mitigate comorbidity-related negative effects. The authors repeatedly emphasized that physical activity training improves mental health. However, this study is a cross-sectional study, but not an intervention study. Therefore, it is not possible to conclude this from the present results. Correlation and causation are different. It is both possible that low activity promotes depression and, conversely, that depression suppresses activity.

Response 3: We appreciate your comment. In accordance with your suggestion, we have made the necessary revisions in discussion on page 13 as follows:

“These results may be attributable to anxiety and depression negatively influencing intention to engage in physical activity in comorbidity patients, resulting in a cycle in which physical inactivity contributed to the patients’ anxiety and depression. This finding should be further explored to improve psychological conditions in patients on hemodialysis with comorbidities.”

Reviewer 4 Report

Thank you for corrections of the manuscript. It is much easier to read now. My major concern related to study aim and design cannot be addressed. It is time to perform interventional studies. 

Author Response

Point 1: Thank you for corrections of the manuscript. It is much easier to read now. My major concern related to study aim and design cannot be addressed. It is time to perform interventional studies..

Response 1: Thank you for your positive comments and encouragement. Regarding the study aim, we have edited the study aim to fit the cross-sectional design in abstract (on page 1).

“This study aims to examine whether physical patterns are associated with anxiety, depression and fatigue among patients with and without comorbidities who are on hemodialysis.”